# Neuroinflammation and Epilepsy: From Pathophysiology to Therapies Based on Repurposing Drugs

**DOI:** 10.3390/ijms25084161

**Published:** 2024-04-09

**Authors:** Pascual Sanz, Teresa Rubio, Maria Adelaida Garcia-Gimeno

**Affiliations:** 1Instituto de Biomedicina de Valencia, Consejo Superior de Investigaciones Científicas, Jaime Roig 11, 46010 Valencia, Spain; trubio@ibv.csic.es; 2Centro de Investigación Biomédica en Red de Enfermedades Raras (CIBERER), 46010 Valencia, Spain; 3Faculty of Health Science, Universidad Europea de Valencia, 46010 Valencia, Spain; 4Department of Biotechnology, Escuela Técnica Superior de Ingeniería Agronómica y del Medio Natural, Universitat Politécnica de València, 46022 Valencia, Spain; magar27m@btc.upv.es

**Keywords:** neuroinflammation, astrocytes, microglia, peripheral immune cells, seizures, epilepsy, repurposing drugs

## Abstract

Neuroinflammation and epilepsy are different pathologies, but, in some cases, they are so closely related that the activation of one of the pathologies leads to the development of the other. In this work, we discuss the three main cell types involved in neuroinflammation, namely (i) reactive astrocytes, (ii) activated microglia, and infiltration of (iii) peripheral immune cells in the central nervous system. Then, we discuss how neuroinflammation and epilepsy are interconnected and describe the use of different repurposing drugs with anti-inflammatory properties that have been shown to have a beneficial effect in different epilepsy models. This review reinforces the idea that compounds designed to alleviate seizures need to target not only the neuroinflammation caused by reactive astrocytes and microglia but also the interaction of these cells with infiltrated peripheral immune cells.

## 1. Introduction

### Neuroinflammation: A General Overview

In general, inflammation is a protective process developed by the organism to maintain its homeostasis. The goal of the inflammatory process is to fight against damage in living cells, by promoting changes to minimize its harmful effect and eventually recovering the homeostatic status. Therefore, inflammation is a positive defensive process to maintain the organism in a healthy state (see [1,2,3] for a review). However, when the cause of the neuroinflammatory reaction is excessive or if it is maintained for a long period of time, inflammation becomes a detrimental process [1,4].

In the brain, neuroinflammation is a response of the innate immune system which is mainly mediated by reactive glia (astrocytes and microglia) that release cytokines, chemokines, reactive oxygen species (ROS), and other proinflammatory mediators [4,5].

The inflammatory process is composed of the following elements:

(1) Inflammatory inducers: These are compounds that initiate inflammatory signaling pathways. We can distinguish between extracellular inducers: pathogen-associated molecular patterns (PAMPs) (e.g., structural elements found within bacterial and fungal cell walls); intracellular inducers, which are compounds produced by stressed, damaged, or malfunctioning cells and tissues, such as damage-associated molecular patterns (DAMPs) (e.g., high-mobility group box 1 protein (HMGB1), histones, adenosine triphosphate (ATP), and reactive oxygen species (ROS)) [6]. In neurodegenerative diseases, inflammation may be triggered by the accumulation of aggregates or modified proteins, which could also be considered DAMPs [7].

Recently, a new group of compounds released by alterations in the homeostatic processes has been defined as HAMPs, homeostasis-altering molecular processes (e.g., changes in intracellular ion levels, modification of the actin cytoskeleton, etc.) [5] (Figure 1).

(2) Sensors: These are pattern recognition receptors (PRRs) that identify the presence of inflammatory inducers. They can be located at the level of the membrane (e.g., Toll-like receptors (TLRs), C-type lectin receptors (CLRs), etc.) or located in the cytosol (e.g., RIG-1-like receptors (RLRs), nucleotide-binding oligomerization domain (NOD), leucine-rich repeat receptors (NLRs), etc.) [5,8]. In [9], we review the components of these pathways.

(3) Effectors: The interaction of inducers with the corresponding sensors triggers the activation of signaling cascades (e.g., mitogen-activated protein kinase (MAPKs) and Akt serine/threonine protein kinase (Akt)) by different mechanisms, which—upon the activation of the intermediate elements of the signaling process—ends with the activation of transcriptional factors involved in the production of proinflammatory mediators (e.g., nuclear factor kappa B (NF-kB) and interferon regulatory factor 3 (IRF3)). One of the main proinflammatory mediators is NF-kB, which has a strategic position at the crossroads between oxidative stress and inflammation. Neuronal dysfunction is closely associated with the activation of NF-kB and the expression of proinflammatory cytokines (see [9] for a review) (Figure 1).

At the same time as the occurrence of the activation of the inflammatory process, a parallel pathway related to the resolution of inflammation is also activated [10,11]. In this process, the activation of PRRs leads to the transcriptional activation of interferon regulatory factors which limit the spread of the insult in the central nervous system (CNS) and diminishes neurodegeneration. Reactive glia are also involved in this pathway by producing anti-inflammatory compounds such as interleukin 10 (IL-10), arginase, chitinase-3 like protein 1 (CHI3L1), and interleukin-1 receptor antagonist protein (IL-1Ra), which are neuroprotective. In this way, the inflammatory pathway is counterbalanced by the resolution of the inflammation process to regulate the extent of inflammation [11]. However, if the inflammatory insult persists, there is a sustained recruitment of inflammatory cells at the site of inflammation because of the reduced clearance of affected cells, which—by releasing DAMPs—sustain an inflammatory cascade [5].

In this work, we describe the general pathophysiology of the neuroinflammatory process, with a special emphasis on the relationship between neuroinflammation and epilepsy. Epilepsy is a neurological disorder characterized by a predisposition to epileptic seizures, and is subject to the associated cognitive, psychological, and social consequences [10,12]. Epilepsy affects around 1% of the total global population and it is caused by acquired insults in the brain (e.g., after stroke or traumatic brain injury), infectious diseases, autoimmune diseases, and genetic mutations [10,12]. At present, despite the availability of many anti-seizure medications (ASMs), approximately one-third of patients with epilepsy do not achieve seizure control or soon become resistant to the effects of the ASMs [13]. Consequently, there is a critical need for the development of innovative anti-epileptogenic treatment strategies to either ameliorate the progression or/and limit the detrimental consequences of the disease. Since there are cases where neuroinflammation and epilepsy are comorbid conditions, here, we summarize some examples of repurposing drugs with anti-inflammatory properties that are beneficial in either animal models or patients with epilepsy. In addition, we discuss the three main cell types involved in neuroinflammation, namely (i) reactive astrocytes, (ii) activated microglia, and infiltration of (iii) peripheral immune cells in the central nervous system, and the possible relationship of these cells in some cases of epilepsy.

## 2. Methodology

This is a narrative review. We used the PubMed database and the following search terms “astrocytes AND neuroinflammation”, “microglia AND neuroinflammation”, “astrocytes AND (seizures OR epilepsy)”, “microglia AND (seizures OR epilepsy)”, “neuroinflammation AND (seizures OR epilepsy)”. We collected outstanding information from the past 5 years (2019–2023), and also critical manuscripts from older years.

## 3. Cell Types of the CNS Involved in Neuroinflammation

It is considered that the main cell types of the CNS involved in the release of proinflammatory mediators are astrocytes, microglia, and infiltrated peripheral immune cells.

### 3.1. Astrocytes

Originally defined as the cells that support the function of neurons in the CNS, astrocytes are currently recognized to play key roles in CNS physiology. On the one hand, astrocytes have housekeeping functions: they supply neurons with energy substrates (e.g., lactate, glutamine); they are involved in the uptake of neurotransmitters (e.g., glutamate) to ensure regular synaptic transmission; they regulate ion homeostasis (e.g., K+ ions) and pH; they play a key role in regulating oxidative stress (e.g., by releasing glutathione); and they release growth factors and other trophic molecules (e.g., brain-derived neurotrophic factor, BDNF; glial-derived neurotrophic factor, GDNF; epidermal growth factor, EGF; transforming growth factor beta, TGF-beta). On the other hand, astrocytes have a key role in the assistance and maintenance of the synapses, being involved in the formation of new synapses as well as in the elimination of non-functional ones by phagocytosis (synaptic pruning) (see [14,15] for a review) (Figure 2).

However, under stress conditions, astrocytes suffer a progressive change in morphology and gene expression by a process called reactive astrogliosis. In this process, astrocytes lose their supportive functions and acquire others related to the expression of characteristic cellular markers (e.g., glial fibrillary acidic protein, GFAP; vimentin) and to the secretion of proinflammatory mediators (e.g., cytokines, chemokines; see below), which could induce neuronal death (see [15,16] for a review). Reactive astrocytes also release lipoproteins containing apolipoprotein E (APOE) and apolipoprotein J (APOJ) proteins and toxic lipids (saturated long-chain fatty acids and very long chain fatty acids), which trigger an endoplasmic reticulum stress response in neurons, with the concomitant activation of the eukaryotic translation initiation factor 2-alpha kinase (PERK), inositol-requiring protein 1 (IRE1), and activating transcription factor 6 (ATF6) signaling pathways and the activation of pro-apoptotic mediators (e.g., CCAAT/enhancer-binding protein homologous protein (CHOP), p53 up-regulated modulator of apoptosis (PUMA) and caspase 3), leading to neuronal cell death by apoptosis [17] (Figure 2).

Due to the loss of their functional properties, reactive astrocytes fail in the maintenance of different homeostatic systems, which could lead to hyperexcitability. They have altered K^+^ and water homeostasis, since they have decreased levels of the inward rectifying channel Kir4.1, which transports K^+^ inside the cells, accompanied by reduced water entry through the aquaporin channel AQP4. This dysfunction leads to excessive extracellular K^+^, which predisposes people to seizures (reviewed in [18,19,20]) (Figure 2).

Reactive astrocytes also have reduced glutamate uptake due to alterations in the expression of the glutamate transporter genes or post-transcriptional and/or post-translational modifications of the corresponding proteins [21]. Due to this dysfunction, there is an increase in the levels of glutamate in the synaptic cleft that activates the glutamate receptors which are present in the post-synaptic neurons (N-Methyl-D-aspartic acid (NMDA), α-amino-3-hidroxi-5-methyl-4-isoxazolpropionic acid (AMPA), Kainate, and metabotropic glutamate (mGLUR) receptors), leading to excitotoxicity. Moreover, reactive astrocytes enhance their expression of these glutamate receptors, which worsens the hyperexcitability [19]. In addition, the release of cytokines such as tumor necrosis factor (TNF), interleukin 6 (IL-6), and interleukin 1 beta (IL-1b) by reactive astrocytes aggravate excitotoxicity since, on the one hand, these molecules stimulate the release of glutamate from these cells, and on the other hand, these cytokines increase the functionality of post-synaptic neuronal glutamate receptors (e.g., by enhancing the phosphorylation and activation of subunits of the NMDA receptor, and by increasing the surface expression of AMPA receptors) (reviewed in [20,22]) (Figure 2). These effects could explain the relationship between reactive astrocytes and hyperexcitability.

Reactive astrocytes have reduced expression of glutamine synthase (reviewed in [23,24]). This enzyme transforms glutamate into glutamine, which is transported to neurons to provide a substrate for the synthesis of glutamate or gamma amino butyric acid (GABA), depending on the type of neuron. If the production of glutamine is reduced, then neurons will suffer a shortage of glutamine that will affect—more importantly—the inhibitory GABAergic neurons, which will not produce GABA, leading to hyperexcitability [20] (Figure 2).

Changes in adenosine metabolism are also produced in reactive astrocytes. These cells increase the expression of adenosine kinase, which converts adenosine into adenosine monophosphate (AMP), in this way lowering the levels of free adenosine, which has anti-seizure properties [13,25]. 

Reactive astrocytes have also dysfunctional energetic metabolism. Due to a decrease in glutamate uptake, there is a decrease in the production of energy from this amino acid by the tricarboxylic acid cycle (TCA), leading to a shortage in energy production and mitochondrial dysfunction. This leads to increased levels of reactive oxygen species (ROS) and oxidative stress, which is detrimental to the neurons [20,21] (Figure 2).

Reactive astrocytes have an activated NF-kB signaling pathway that ends with the expression of proinflammatory mediators such as cytokines (e.g., TNF, IL-6, IL-1b, Lipocalin 2), chemokines (e.g., CCL2, CCL5, CXCL10), and components of the complement cascade (e.g., C1q, C3), which enhances synaptic pruning and neuronal cell death [26,27]. On the other hand, astrocytes are the target of different inflammatory ligands that act on specific pattern recognition receptors (PRRs), such as Toll-like receptors (TLRs) and NOD-type receptors (NLRs). In relation to epilepsy, the most important ones are IL-1R1 (receptor of IL-1b), TLR2, TLR3, and TLR4 (receptors of HMGB1 and LPS), which—through the activation of NF-kB—activate the expression of inducible nitric oxide synthase (iNOS), cyclooxygenase 2 (COX2), IL-6, TNF, CCL2, and CXCL10, among other proinflammatory mediators (reviewed in [20,28]) (Figure 2). 

At the same time, astrocytes produce neuroprotective molecules such as the IL-1R1 antagonist IL-1Ra and anti-inflammatory cytokines (IL-4, IL-13, and IL-10) [29]. They also produce microRNAs (miRNAs) associated with a downregulation of the TLR signaling pathways (miRNA-146a and miRNA-147b), leading to an anti-inflammatory effect [28]. This may sound counterintuitive, but in this way, the initial inflammatory process can be balanced to recover homeostasis.

In summary, the presence of metabolic failure, oxidative stress, excitotoxicity, and inflammation—as a consequence of astrocyte reactivity—contributes to the appearance of hyperexcitability. Recently, an astrocytic basis for epilepsy has been proposed, and results in both animal models and human samples indicate that astrocyte dysfunction can participate in hyperexcitation, neurotoxicity, and seizure spreading [30]. Perhaps this is the reason why the European Commission of the International League Against Epilepsy (ILAE) recognized the role that glia and inflammation may have in the development of seizures and epileptogenesis as a top research priority, and encouraged the identification of glial targets as a basis for the development of more specific anti-seizure medications (ASMs) [31]. This reinforces the role of astrocytes as epileptogenic drivers in acquired epilepsies [20,32].

### 3.2. Microglia

Microglia are the main component of the innate immune system in the CNS. They patrol around the CNS and are the first to respond to even small changes that affect the CNS. They are activated by different conditions such as the release of ATP from neuronal death cells or the presence of high levels of glutamate, among other insults [33]. This activation leads to a change in their morphology, where they become more rounded and less ramified [34].

In general, microglial activation occurs in advance of astrocyte reactivity [35], although there are some cases where astrocytes become reactive in the absence of activated microglia [36]. Microglia release IL1a, TNF, and C1q, which induce the acquisition of the reactive phenotype in the astrocytes [26,27]. In addition, microglial TNF induces the expression of vascular cell adhesion molecule-1 (VCAM1) and intercellular adhesion molecule-1 (ICAM1) in endothelial cells, favoring the infiltration of peripheral immune cells, which—by the production of more cytokines and chemokines—aggravates neuroinflammation (see below) [37].

The main functions of microglia in the CNS are the release of cytokines and chemokines, the phagocytosis of apoptotic cells, and the control of synapsis; in addition, they modulate neuronal activity and maintain a cross-talk with astrocytes (see below) [35,38,39]. Upon activation, microglia acquire a neurotoxic profile and release nitric oxide (NO), ROS, cytokines (IL-1b, IL-18, IL-6, TNF, IL-1a), chemokines (CCL2, CCL5), complement proteins (C1q), and proinflammatory miRNAs (e.g., miRNA155). However, under moderate activation, microglia have a neuroprotective role by the release of anti-inflammatory mediators, such as Arg1, IL-4, IL-10, TGF-beta, CCL17, interferon growth factor 1 (IGF-1), GDNF, BDNF, platelet-derived growth factor (PDGF), and vascular endothelial growth factor (VEGF) (reviewed in [29,40,41]).

Microglia plays a main role in the regulation of synaptic plasticity and neurogenesis. In conjunction with astrocytes, they participate in the removal of weaker synapses. They recognize complement molecules such as C3 and C1q, which are located on the surface of affected synapsis, acting as “eat-me” signals. Under inflammatory conditions, there is an increase in the levels of C3 and C1q, which leads to an increase in synaptic pruning, which is associated with neurodegeneration [42]. In patients with epilepsy, an increase in the levels of C1q in affected areas has been reported (see [43,44,45], for a review).

### 3.3. Microglia–Astrocyte Crosstalk

As indicated above, there is a bidirectional communication between microglia and astrocytes [39]. Microglia induce astrocyte activation and determine the fate of astrocytes, and astrocytes may trigger microglial activation and control their cellular function (reviewed in [34,46]).

On the one hand, microglia secrete factors that affect astrocyte physiology. Upon acute neuronal hyperactivity and activation of glutamate–NMDA receptors, neurons release ATP, which stimulates microglial purinergic receptors (P2Y12) and triggers the secretion of proinflammatory inducers (e.g., IL-1b, IL-6, TNF, NO), which affect the immune functions of the astrocytes. These inducers are sensed by pattern recognition receptors (PRRs) in the astrocytes, which promote the production of inflammatory factors (cytokines, chemokines, and complement components), which amplify the inflammatory response [46,47]. One of the best-known examples of the microglia-derived activation of astrocytes is the release of TNF, IL-1a, and C1q, which induce the astrocytic neurotoxic phenotype [26,27]. Microglia also negatively affects the expression and activity of astrocytic glutamate transporters (GLT-1 and GLAST) [46]. Moreover, microglia induce the release of glutamate from the astrocytes, worsening the excitotoxicity produced by the accumulation of glutamate in the synaptic cleft [48]. At the same time, microglia also secrete anti-inflammatory compounds (TGF-alpha, IL-4, IL-10) which alleviate astrocyte reactivity [48] (Figure 3).

On the other hand, astrocytes secrete factors that affect microglia physiology. Astrocytes produce lipocalin-2 (Lcn2), which is a critical contributor to the inflammatory activation of astrocytes and enhances microglial activity under inflammatory and pathological conditions of the CNS [46]. Astrocytes also produce glial-derived neurotrophic factor (GDNF), which increases microglial activation and neuroinflammation. In addition, astrocytes regulate microglial migration and phagocytosis; they produce complement factor C3, which regulates microglial phagocytosis, and the chemokines CCL2 and CXCL10 which recruit microglia to the damaged area to eliminate damaged cells and cellular debris [41,46]. In parallel, astrocytes also release anti-inflammatory mediators such as orosomucoid-2 (ORM2), and TGF-beta, which prevent microglial activation [48] (Figure 3).

In several forms of epilepsy, there is a higher reactivity of microglia in the cornus ammonis CA3 and CA1 areas of the hippocampus, which is associated with higher neuronal death and hyperexcitability (reviewed in [38,49,50]). It has been assumed that microglia are activated rapidly after a seizure. Activated microglia, on the one hand, releases proinflammatory cytokines and toxic compounds (e.g., reactive oxygen species, ROS; nitric oxide, NO); on the other hand, they have dysfunctional phagocytosis and defective proteostasis, which is detrimental for neuronal function (see [51,52,53] for a review). At the same time, microglia induce the subsequent activation of astrocytes, which develops slowly but is maintained with time, contributing to the pathology of epilepsy by the mechanisms described above [53]. In addition to this sequence of events, it has also been reported that astrocytes, per se, can drive seizures without the participation of activated microglia [36].

### 3.4. Peripheral Immune Cells Infiltration into the CNS

The CNS is no longer an immune-privileged organ since there is a substantial crosstalk between cells of the central and the peripheral immune system. Although inflammation in the CNS starts with the activation of microglia and/or astrocytes, treatments aimed at suppressing microglia/astrocyte activation are not enough to prevent neuroinflammation; thus, a more comprehensive approach that includes the peripheral immune system is necessary [54]. Cells from the peripheral immune system release cytokines which can cross the blood–brain barrier (BBB) to cause neurotoxicity and to activate microglia and astrocytes. In addition, dysregulated signaling between astrocytes, endothelial cells, and pericytes may cause the BBB to become permeable, allowing the invasion of peripheral immune cells into the CNS; they can gain access to the brain by diapedesis once a recruitment signal (e.g., release of chemokines) has been produced. It is important to note that infiltration can occur without an overt opening of the BBB (see [54,55,56] for a review).

The main peripheral immune cells that infiltrate the brain are neutrophils, monocytes/macrophages, B-cells, and T-cells. A temporal trend in the infiltration of peripheral immune cells in the brain has been described, in which neutrophils seem to be the first and then they dissipate. The second type of cells is monocytes/macrophages; they are recruited by the release of cytokines/chemokines by the astrocytic endfeet, endothelial cells, and pericytes. The chemokine CCL2 is a potent chemoattractant for cells of the monocytic lineage, which express the corresponding CCR2 receptor. The activation of this receptor induces a signal transducer and an activator of transcription 3 (STAT3)-dependent transcriptional activation of proinflammatory mediators, aggravating neuroinflammation. The third type of cells are the T-cells, which last longer in the brain parenchyma [57]; among the T-cells, the most abundant ones are CD4+ (T-helper Th cells, comprising proinflammatory Th-1 cells, anti-inflammatory Th-2 cells (Th17)—involved in the recruitment of neutrophils—and Treg cells, playing a role in immunosuppression), CD8+ cytotoxic T cells, involved in destroying damaged cells, and gamma delta T-cells (see [37,54,58] for a review). T-cells are attracted to the endothelium by the presence of chemokines (CCL2 and CCL5), which interact with the CCR2 and CCR5 receptors present in the T-cells. Then, T-cells attach to the endothelium through a combination of specific adhesion molecules named selectins (P-selectin, VCAM1, ICAM1)—in the case of endothelial cells—and specific selectin ligands that are expressed in T cells. Then, T-cells produce interferon-gamma (IFNg), which induces the production of proinflammatory mediators by macrophages, and also the chemokine CXCL10, which enhances the recruitment of Th-1 cells [37,58]. In the case of cytotoxic CD8+ T cells, by recognizing specific antigens, they release granzymes and perforins, forming pores on the membranes of target cells and inducing apoptosis. The presence of CD8+ T cells is considered a worse outcome of the neuroinflammatory disease [37,58]. 

In relation to epilepsy, it has been reported that the infiltration of neutrophils is involved in the induction of acute brain inflammation after status epilepticus, and seizure frequency correlates with the number of infiltrating monocytes [56]. Infiltration of CD8+ T-cells has been detected in epileptogenic areas, and it is considered that the greater infiltration of CD8+ T-cells in the CA1 region of the hippocampus shows a greater positive correlation with neuronal loss in the area (reviewed in [59,60]) (Figure 4).

## 4. Neuroinflammation and Epilepsy

As has been described above, there is a direct relationship between neuroinflammation and epilepsy: brain inflammation has a role in the etiopathogenesis of seizures, and seizures cause neuroinflammation (reviewed in [10,61,62]). This notion is supported by different pieces of evidence: (i) inflammation is induced by recurrent seizures; (ii) the release of proinflammatory cytokines contributes to cell loss; (iii) seizure-induced brain inflammation is long-lasting and can persist for days; (iv) inflammation precedes the onset of spontaneous seizures, suggesting that uncontrolled inflammation may contribute to the development of the epileptic process; (v) treatment with specific anti-inflammatory agents reduces experimental seizures [10,61,62].

One form of triggering inflammation is the release of DAMPs (ATP, HMGB1, other compounds) from the dead cells due to seizures. These mediators trigger the inflammatory response in microglia and astrocytes in the brain parenchyma; then, the infiltration of peripheral immune cells may trigger a sustained inflammatory cascade [63]. Neuroinflammation may also occur without neuronal cell loss because of enhanced neuronal activation (neurogenic inflammation) [64,65,66] (Figure 5).

On the contrary, neuroinflammation may occur before the onset of epilepsy: dysregulation of the glial immune–inflammatory function may predispose or directly contribute to the generation of seizures. As indicated above, astrocyte reactivity leads to different changes in the homeostasis of the glutamate transporters, which leads to hyperexcitability, excitotoxicity, and—eventually—the appearance of seizures [25,62] (Figure 5). For this reason, the use of anti-inflammatory drugs to reduce astrocyte reactivity may have disease-modifying effects in epilepsy. In this way, modulation of specific inflammatory pathways could be a new therapeutic approach for pharmaco-resistant focal epilepsies [3,63].

## 5. Use of Repurposing Drugs as an Anti-Inflammatory Therapeutic Approach in Epilepsy

Several compounds that target different inflammatory signaling pathways have demonstrated their efficacy in epilepsy models. In ref. [9], we collected some of these compounds: in brief, we defined the beneficial effects of the following: of anakinra and anti-IL-1b monoclonal antibodies (Mabs) in inhibiting the IL1-b/IL-1R1 axis; of BoxA, P5779, and anti-HMGB1 Mabs in inhibiting the TLR4/RAGE axis; of anti-TNF Mabs, dihydrothalidomide, Nilotinib, and cannabinoids in inhibiting the TNF/TNFR1 axis; of anti-IL-6 Mabs and WP1066 in inhibiting the IL-6/Jak-STAT axis; of losartan in inhibiting the TGFbeta axis; of anti-CXCL10 Mabs in inhibiting the CXCL10/CXCR3 axis; and of anti-C1q Mabs and NLY01 in inhibiting microglia activation.

In this work, we focus our attention on some repurposed drugs with anti-inflammatory effects that display a beneficial effect in epilepsy models (see Table 1).

### 5.1. Metformin

Metformin belongs to the family of biguanide compounds that have glucose-lowering effects. This compound is currently the most commonly prescribed drug for type 2 diabetes (T2D) and is taken by an estimated 150 million people worldwide. Due to its superior safety profile, it has become the first-line treatment for T2D and is now featured on the World Health Organization’s essential medicines list (reviewed in [67,68]). However, metformin has additional benefits on top of its use as an anti-diabetic drug: it is effective in the treatment of multiple diseases, delays the aging process, and alleviates inflammation [67,68]. Recently, the beneficial effects of metformin have also been reported for different neurodegenerative diseases, such as Alzheimer’s [69], Parkinson’s [70], and Huntington’s diseases [71], as well as multiple sclerosis [72], among others [73,74]. Metformin has also been shown to be able to attenuate the generation of seizures by delaying the onset of epilepsy, reducing neuronal loss in the hippocampus, and preventing cognitive impairments in both acute and chronic epilepsy models. Its anti-seizure effects could be attributed to both AMPK-dependent and AMPK-independent mechanisms [75,76].

We used metformin in a mouse model of Lafora disease, a devastating form of progressive myoclonus epilepsy, and showed that it reduces some of the hallmarks of the disease, such as the accumulation of polyglucosans and polyubiquitin aggregates in the brain, and reactive astrogliosis, resulting in improvements in the results of neuropsychological tests [77]. In addition, metformin decreases susceptibility to seizures, reduces the number and length of seizures, and eliminates the mortality induced by the pro-convulsive agent pentylenetetrazol (PTZ) in Lafora disease mouse models [78]. These results allowed the designation of metformin as an orphan drug for the treatment of Lafora disease by the European Medicines Agency (EMA) in 2016 and the United States Food and Drug Administration (FDA) in 2017. Importantly, metformin has been used to treat a group of LD patients and showed a beneficial effect by slowing down the progression of the disease [79].

### 5.2. Fingolimod

Sphingosine-containing phospholipids are prominent signaling molecules in the CNS. One of them, sphingosine-1-phosphate (S1P), signals through a family of G-protein-coupled receptors (S1PR1-5), leading to cytokine production. Fingolimod (FGD) is an antagonist of S1PRs. Due to its immunosuppressant effects, in 2010, the FDA approved its use for the treatment of multiple sclerosis. FGD has multiple actions at the level of the central nervous system (CNS): (i) in endothelial cells, it reduces the permeability of the blood–brain barrier (BBB) by decreasing the expression of ICAM-1 and reducing the binding of leukocytes to endothelial cells; (ii) in neurons, it protects from excitotoxic cell death, preventing apoptosis; (iii) in astrocytes, it inhibits the production of proinflammatory cytokines, chemokines, and neurotoxic substances (e.g., IL-6, COX2, VEGF); it also increases the production of neuroprotective factors; (iv) in microglia, it reduces microglia activation, and also reduces the production of proinflammatory cytokines (IL-6); (v) in oligodendrocytes, it promotes the renewal of oligodendrocytes and enhances remyelination. In conclusion, fingolimod reduces inflammation, excitotoxicity, glial activation, and BBB destruction, and improves neurogenesis [80]. In addition, it upregulates the production of neurotrophic factors such as BDNF and reduces the infiltration of T-lymphocytes into the brain parenchyma [81]. 

FGD’s beneficial effects on different neurological disorders such as stroke [80], hypoxia [82], and epilepsy have been described (reviewed in [83,84,85,86]). Recently, we have described the beneficial effects of FGD in a mouse model of Lafora disease, a particular type of progressive myoclonus epilepsy. In this model, FGD reduced reactive astrogliosis-derived neuroinflammation and T-lymphocyte infiltration, which correlated with an improved behavioral performance among the treated animals [87]. However, in the case of the Rett syndrome, although the administration of fingolimod was safe in children with this disorder, it did not provide supportive evidence for an effect on clinical, laboratory, and imaging measures in these patients [88].

### 5.3. Dimethyl Fumarate

Dimethyl fumarate (DMF) is an immunomodulatory drug approved by the FDA in 2013 for the treatment of autoimmune diseases such as multiple sclerosis. It has been reported that DMF activates the nuclear factor erythroid 2-related factor (Nrf2), having antioxidant effects [89,90]. In addition, DMF reduced the infiltration of peripheral immune cells into the spinal cord in a mouse model of experimental autoimmune encephalitis (EAE) and in patients with multiple sclerosis [91,92]. DMF also reduces microglial activation (Iba1) and the infiltration of CD4+ and CD8+ T-lymphocytes in the brain, as shown by a rat model of EAE [93]. Recently, it has been described that DMF induces robust anti-inflammatory signaling by activating HCAR2, a G-protein-coupled membrane receptor expressed in immune cells [89]. Related to epilepsy, a beneficial effect of DMF in alleviating seizures in a pentylenetetrazole (PTZ)-induced rat model has been reported [94,95]. In addition, it has been reported that administration of DMF following status epilepticus increased Nrf2 activity, attenuated the status of epilepticus-induced neuronal cell death, and decreased seizure frequency and the total number of seizures compared to vehicle-treated animals [96]. However, when we used DMF in the Lafora disease mouse model, we observed only a minor effect; it was less effective in preventing neuroinflammation and T-lymphocyte infiltration [87].

### 5.4. Propranolol

Propranolol is a β-adrenergic antagonist that is used in the treatment of high blood pressure since it improves blood flow and reduces the strain on the heart. This compound can ameliorate microglial reactivity [97,98]. Perhaps this is the reason for its recognized anti-neuroinflammatory [99,100] and neuroprotective properties [101]; but its molecular mechanism is still unknown. Recently, we described the beneficial effect of propranolol on a mouse model of Lafora disease. Propranolol improved not only the attention defects but also neuronal disorganization, astrogliosis, and microgliosis that were present in the hippocampus of the mice in this model [102].

**Table 1 ijms-25-04161-t001:** Primary and secondary indications of the repurposed drugs described in this work. The mechanism of action in the primary and secondary indication, and the references related to the secondary indication are indicated.

Compound	Primary Indication	Primary Mechanism of Action	Secondary Indication	Secondary Mechanism of Action	References of Secondary Indication
Metformin	Type 2 Diabetes	Reduces hepatic glucose production	Anti-seizure effect	Anti-inflammatory, with AMPK-dependent and -independent effects	Animal models: [75,76,77,78] Humans: [79]
Fingolimod	Multiple sclerosis	Antagonist of sphingosine-1-phosphate receptors with broad action at the central nervous system	Anti-seizure effect	Anti-inflammatory; prevents leukocyte infiltration into the brain parenchyma	Animal models: [83,84,85,86,87] Humans: [88]
Dimethyl Fumarate	Multiple sclerosis	Antioxidant; improves erythroid 2-related factor-dependent action	Anti-seizure effect	Anti-inflammatory; prevents leukocyte infiltration into the brain parenchyma	Animal models: [94,95,96]
Propranolol	b-adrenergic antagonist	Treatment of high blood pressure; it improves blood flow and reduces the strain on the heart	Anti-seizure effect	Anti-inflammatory; reduces microglial activation	Animal models: [102]
Ibuprofen	Non-steroidal anti-inflammatory drug	Inhibitor of cyclooxygenase 2	Anti-seizure effect	Reduces activity of post-synaptic NMDA glutamate receptors	Animal models: [103,104,105]
N-acetyl Cysteine	Antioxidant	Precursor of reduced glutathione	Anti-seizure effect	Antioxidant, scavenger of reactive oxygen species	Animal models: [84] Humans: [106]

### 5.5. Ibuprofen

Ibuprofen belongs to the group of non-steroidal anti-inflammatory drugs (NSAIDs). It is an inhibitor of cyclooxygenase 2 (Cox2) that reduces the activation of the NLRP3 inflammasome and the release of IL-1b and IL-18. In addition, it negatively regulates the function of the NMDA post-synaptic glutamate receptors, in this way decreasing hyperexcitability and the appearance of seizures in a (PTZ)-induced rat model [103,104]. Ibuprofen also reduced sensitivity to PTZ in a Lafora disease mouse model, decreasing the severity and duration of seizures and downregulating the expression of proinflammatory mediators [105].

### 5.6. N-Acetyl-Cysteine

N-acetyl-cysteine (NAC) is an antioxidant compound and a precursor of reduced glutathione (GSH). It has been reported that the administration of NAC is beneficial for patients suffering from progressive myoclonus epilepsy type I (Unverricht-Lundborg disease) since they showed marked improvement in seizures, ataxia, and the blockade of symptoms progression [84,106].

Any of these repurposing compounds could be used under the notion of “network pharmacology”, formulated recently, where the design of chosen repurposing drug cocktails could be used as anti-seizure medications. This is because it has been reported that the combined action of different repurposing drugs is more beneficial than the action of a single compound [84].

## 6. Conclusions

There is a close relationship between neuroinflammation and epilepsy. On the one hand, seizures can trigger an inflammatory response in the brain; on the other hand, neuroinflammation affects the regular physiology of astrocytes and microglia, which triggers alterations in different systems (e.g., increased levels of glutamate due to the release of the neurotransmitter from astrocytes, the reduction in the glutamate uptake capacity, etc.), which eventually produces hyperexcitability and excitotoxicity. These lead to the appearance of seizures. We would like to emphasize the important role that the infiltration of peripheral immune cells may play in neuroinflammation and epilepsy. For this reason, compounds designed to alleviate these diseases need to target not only the neuroinflammation caused by reactive astrocytes and microglia but also the functional interaction of these cells with infiltrated peripheral immune cells. The connection between neuroinflammation and epilepsy allows the design of specific anti-inflammatory drugs that could be useful as alternative therapeutics in the treatment of epilepsy, either alone or in combination with general anti-seizure medications.

## Figures and Tables

**Figure 1 ijms-25-04161-f001:**
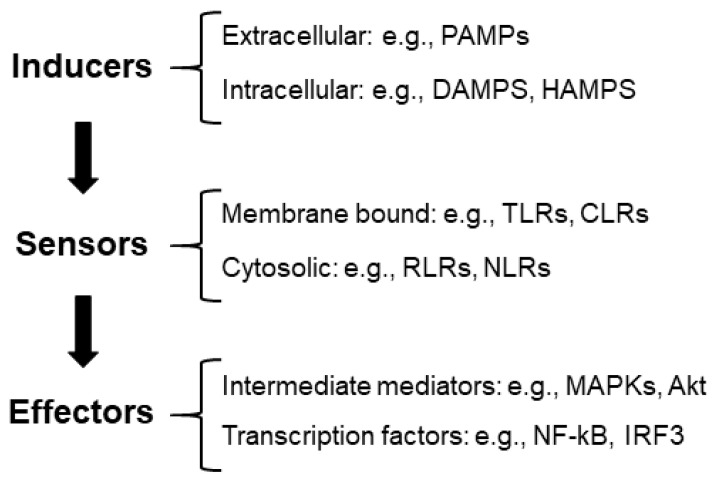
Diagram of the components of the inflammatory reaction. See the text for the description of the different abbreviations.

**Figure 2 ijms-25-04161-f002:**
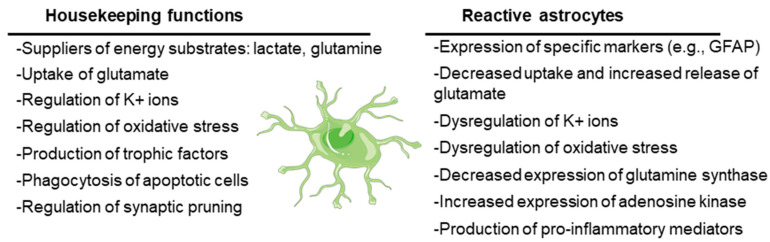
Main housekeeping functions of astrocytes and dysfunctions when they become reactive. The cartoon refers to a healthy astrocyte. See text for details.

**Figure 3 ijms-25-04161-f003:**
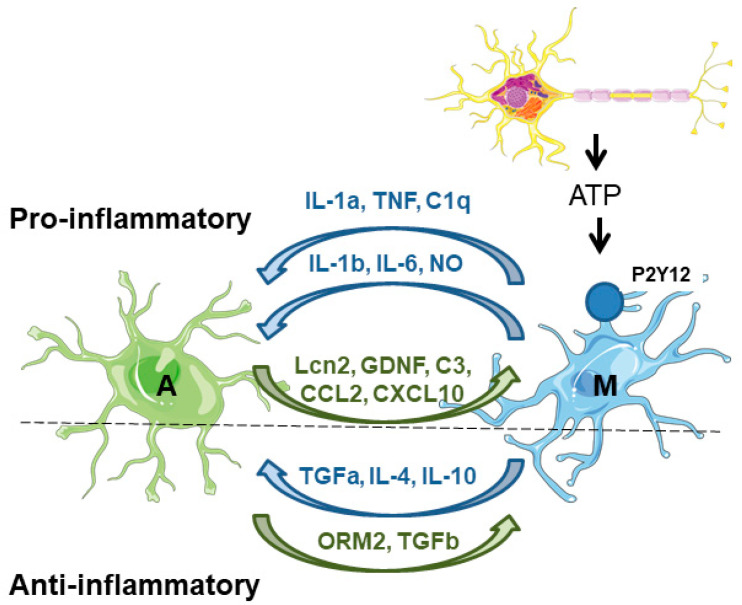
Astrocyte–microglia crosstalk. On the proinflammatory side of the coin, microglia are activated by the release of different components (e.g., ATP) from neuronal cells. Then, microglia induce the activation of astrocytes by the release of different compounds (IL1a, TNF, C1q). In addition, microglia also release proinflammatory mediators (IL-1b, IL-6, and NO) that maintain astrocyte reactivity. Astrocytes respond to microglia by releasing different proinflammatory mediators (Lcn2, GDNF, C3, CCL2, CXCL10) which maintain microglia in their activated state. On the anti-inflammatory side of the coin, microglia release different mediators (TGF-alpha, IL-4, and IL-10) to counteract the excessive activation of the astrocytes. These last cells also produce anti-inflammatory mediators (ORM2, TGF-beta) to reduce microglial activation. A—astrocytes; M—microglia. See text for details.

**Figure 4 ijms-25-04161-f004:**
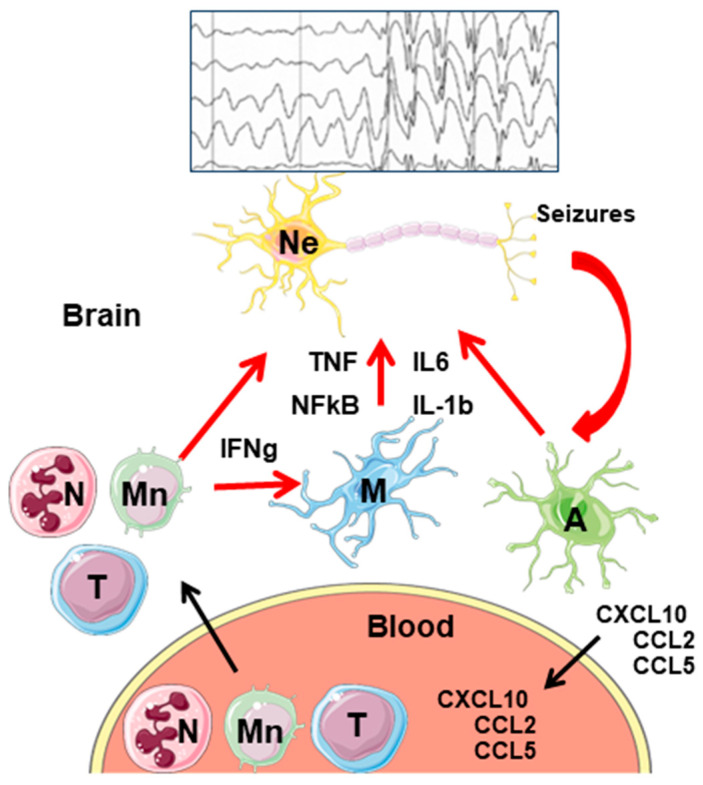
General model of neuroinflammation in epilepsy. Activation of microglia and astrocytes by hyperactive neurons leads to the release of proinflammatory mediators (TNF, IL-6, IL-1b), and the activation of the NF-kB signaling pathway. In addition, astrocytes release chemokines (CXCL10, CCL2, CCL5) that make their way to the blood. There, they attract different peripheral immune cells (neutrophils, monocytes, and T-lymphocytes) which infiltrate into the brain parenchyma. There, they express different proinflammatory mediators that, on the one hand, enhance the proinflammatory properties of astrocytes and microglia; on the other hand, they worsen the proinflammatory landscape. Eventually, neurons die and release DAMPs, which start the cycle again. Ne—neurons; A—astrocytes; M—microglia; N—neutrophils; Mn—monocytes; T—T-lymphocytes. A simplified seizure–spikes diagram is shown.

**Figure 5 ijms-25-04161-f005:**
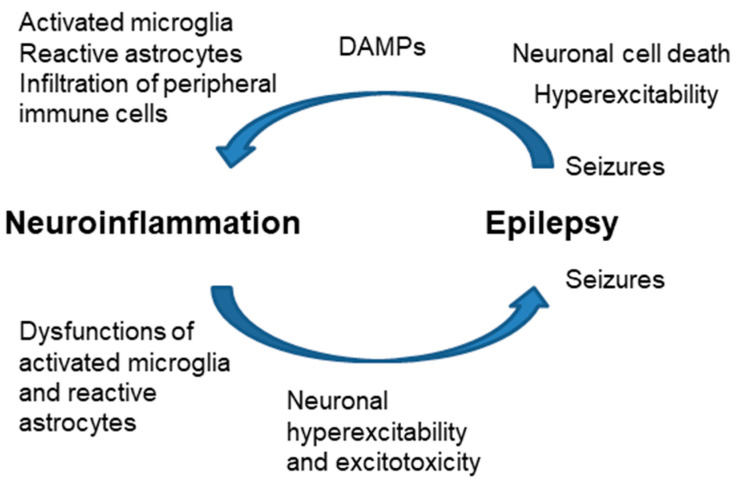
Close relationship of neuroinflammation and epilepsy. Increased neuroinflammation produces dysfunctions in microglia and astrocytes, which end with neuronal hyperexcitability and excitotoxicity, leading to the appearance of seizures. Hyperexcitability and neuronal cell death after seizures release DAMPs (ATP, HMGB1) that activate microglia, and astrocytes and eventually produce the infiltration of peripheral immune cells.

## Data Availability

Data are available upon request.

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
