# Peer review of "Neuroinflammation and Epilepsy: From Pathophysiology to Therapies Based on Repurposing Drugs"

_ijms, 2024, doi:10.3390/ijms25084161_

Round 1

Reviewer 1 Report

Comments and Suggestions for Authors

Neuroinflammation and Epilepsy: from Pathophysiology to Therapies based on Repurposing Drugs

The current manuscript is a review on neuroinflammation and epilepsy with a particular focus on the utility of repurposed molecules.

I have some comments that would strengthen the article:

General comment:

There is no information on how the review was conducted. The authors should consider adding a methodology section to the manuscript to indicate the type of review i.e., scoping or narrative. I understand that the article is not a systematic review, but some information should be added regarding the methodology on how the current review was conducted i.e., search terms used, databases searched, how the articles mentioned in the different sections were identified etc. A figure (like a PRISMA diagram) might also be useful.

Abstract:

Line 13: the three main cell types need to be highlighted: It might be worth rephrasing this as follows:

In this work, we discuss the three main cell types involved in neuroinflammation, namely (I) reactive astrocytes, (II) activated microglia, and infiltration of (III) peripheral immune cells in the central nervous system. We then…[…]

Introduction:

While the introduction describes elements of neuroinflammation, there is not much information on the different aspects to do with epilepsy, which should form a central component of the review. The authors need to add some information on epilepsy and then go onto describing the key evidence how epileptic seizures might be associated with neuroinflammation. Clear distinctions between studies done in animal models versus those done in humans should also be emphasised whenever necessary.

Line 30: It is unclear what is meant by “action of the intruder” and this ought to be modified or removed.

It would be helpful for readers if all abbreviations could be described in their entirety when first mentioned i.e., ATP, ROS (line 43), CHI3L1, and IL‐1Ra (line 68), GFAP (line 96), APOE, APOJ (line 99), PERK, IRE1, ATF6, CHOP, PUMA etc. etc. (line 101-102). Alternatively, an abbreviation section could be adding at the beginning of the manuscript.

“etc”, should be removed [line 43].

Line 74: Please consider adding some objectives and what the key learning outcomes of the review will be. If the focus of the review is to describe the three main cell types, then this should be indicated here.

Methodology:

The review is missing a methods section (see general comment above) and this section ought to be placed here before the main aspects of the review are mentioned.

Section 2.1: Line 90: Please describe what “neuronal synapsis”, is. Otherwise, please check if this ought to be synapses i.e., formation of new synapses.

Figure 1: Is the image of the astrocyte in the housekeeping or reactive state? Please can this be stated in Figure 1.

Line 107 to 112: This paragraph comes across too vague especially when the authors state that “Due to the loss of their functional properties, reactive astrocytes are associated with the appearance of seizures.” The authors should provide further information how the loss of reactive astrocyte functionality leads to the appearance of seizures. Are these seizures categorised to a specific type i.e., focal, generalized? Are prominent to specific regions of the brain?

Line 133: When describing the effect of adenosine metabolism in line 133, please can the authors clarify what is meant by ‘They” in “They increase”? Does “they” reflect changes in adenosine metabolism in reactive astrocytes?

Line 160: more explanation should be added on the role of astrocytes as epileptogenic drivers and cite the relevant empirical evidence (as necessary).

Figure 2: Line 213 – please clarify what is meant by “neuronal dead cells”. From my understanding, the development and progression of epilepsy may not entirely be caused by neuronal cell death, for example, some paediatric neurodevelopmental disorders have co-occurring epilepsy with no neuronal cell death (neurodegeneration). Please also check whether the arrow(s) in Figure 2 ought to be bidirectional. As it stands, it is difficult to discern from Figure 2, how activated astrocytes lead to seizures. Please also described what ‘A’ and ‘M’ means.

Line 234: If microglia can become activated after a seizure, please also explain in this section how reactive/activated microglia may also precipitate seizure(s). Can any information be provided whether reactive/activated microglia have a role in the pre-ictal stage and whether this might lead to the ictal phase and then the post-ictal phase including more deleterious post ictal aspects such as Post-ictal Generalized Electroencephalographic Suppression (PGES)? This can be linked in with the text on line 317 where the authors suggest that neuroinflammation may occur before the onset of seizures.

Line 318: this is an interesting point and should be expanded upon. Can the authors shed light on any instances in which there is a susceptibility to dysregulated glial immune-inflammatory function that could predispose certain individuals to be at a higher risk of seizures?

Section 4: please say what Mabs and box are. I feel a table stating the main repurposed molecules and their putative roles would be useful here. It should also be clearly stated whether they would be useful for the management of epilepsy. Demarcation between studies done in animal models and humans should be clearly indicated.

The potential limitation of these molecules should be also alluded too. For example, the failure of Fingolimod in Rett Syndrome, a paediatric neurodevelopmental disorder, where seizures are common.

Comments on the Quality of English Language

Needs some improvements (see suggestions for authors)

Author Response

The current manuscript is a review on neuroinflammation and epilepsy with a particular focus on the utility of repurposed molecules.

I have some comments that would strengthen the article:

General comment:

There is no information on how the review was conducted. The authors should consider adding a methodology section to the manuscript to indicate the type of review i.e., scoping or narrative. I understand that the article is not a systematic review, but some information should be added regarding the methodology on how the current review was conducted i.e., search terms used, databases searched, how the articles mentioned in the different sections were identified etc. A figure (like a PRISMA diagram) might also be useful.

We thank the reviewer very much for this comment. We would like to indicate that this is a narrative review and not a systematic one. In the new version of the manuscript, we have included a section that describes the way we collected all the information necessary to write this review. Basically, we use PUBMED and the following search terms “astrocytes AND neuroinflammation”, “microglia AND neuroinflammation”, “astrocytes AND (seizures OR epilepsy)”, “microglia AND (seizures OR epilepsy)”, “neuroinflammation AND “seizures OR epilepsy”. We collected the outstanding information from the past 5 years (2019-2023), and also critical manuscripts in older years.

Abstract:

Line 13: the three main cell types need to be highlighted: It might be worth rephrasing this as follows:

In this work, we discuss the three main cell types involved in neuroinflammation, namely (I) reactive astrocytes, (II) activated microglia, and infiltration of (III) peripheral immune cells in the central nervous system. We then…[…]

Following the reviewer’s suggestion we have included the following paragraph in the abstract section: “In this work, we discuss the three main cell types involved in neuroinflammation, namely (i) reactive astrocytes, (ii) activated microglia and infiltration of (iii) peripheral immune cells in the central nervous system. We then…

Introduction:

While the introduction describes elements of neuroinflammation, there is not much information on the different aspects to do with epilepsy, which should form a central component of the review. The authors need to add some information on epilepsy and then go onto describing the key evidence how epileptic seizures might be associated with neuroinflammation. Clear distinctions between studies done in animal models versus those done in humans should also be emphasised whenever necessary.

To address the reviewer’s comment, we have included the following paragraph at the end of the Introduction section. “In this work, we describe the general pathophysiology of the neuroinflammatory process with a special emphasis on the relationship between neuroinflammation and epilepsy. Epilepsy is a neurological disorder characterized by a predisposition to generate epileptic seizures and the associated cognitive, psychological, and social consequences (Devinsky 2018), (Vezzani, 2019). Epilepsy affects around 1% of the total world population and it is caused by acquired insults in the brain (e.g., after stroke or traumatic brain injury), infectious diseases, autoimmune diseases, and genetic mutations (Devinsky 2018), (Vezzani, 2019). At present, despite the availability of many anti-seizure medications (ASMs), approximately one-third of epileptic patients fail to achieve seizure control or soon become resistant to the effects of ASMs (Patel 2019). Consequently, there is a critical need for the development of innovative anti-epileptogenic treatment strategies to either ameliorate the progression or/and limit the detrimental consequences of the disease. Since there are cases where neuroinflammation and epilepsy go together, here we summarize some examples of repurposing drugs with anti-inflammatory properties which are beneficial in either animal models or epileptic patients”.

            In addition, we have included a new Table I where we indicate the action of selected repurposed drugs on animal models and human patients.

Line 30: It is unclear what is meant by “action of the intruder” and this ought to be modified or removed.

Following the reviewer’s suggestion we have rephrased the sentence by saying: “However, when the cause of the neuroinflammatory reaction is excessive or …”

It would be helpful for readers if all abbreviations could be described in their entirety when first mentioned i.e., ATP, ROS (line 43), CHI3L1, and IL‐1Ra (line 68), GFAP (line 96), APOE, APOJ (line 99), PERK, IRE1, ATF6, CHOP, PUMA etc. etc. (line 101-102). Alternatively, an abbreviation section could be adding at the beginning of the manuscript.

We apologize for this mistake. We have indicated the full name of the abbreviation when it appears for the first time in the text.

“etc”, should be removed [line 43].

We have removed the term “etc” from the sentence.

Line 74: Please consider adding some objectives and what the key learning outcomes of the review will be. If the focus of the review is to describe the three main cell types, then this should be indicated here.

On top of what we have modified at the end of the Introduction (see above), we have added: “In addition, we discuss the three main cell types involved in neuroinflammation, namely (i) reactive astrocytes, (ii) activated microglia and infiltration of (iii) peripheral immune cells in the central nervous system, and the possible relationship of these cells in some cases of epilepsy.

Methodology:

The review is missing a methods section (see general comment above) and this section ought to be placed here before the main aspects of the review are mentioned.

In the new version of the manuscript, we have included a Methodology section that describes the way we collected all the information necessary to write this review. Basically, we use PUBMED and the following search terms “astrocytes AND neuroinflammation”, “microglia AND neuroinflammation”, “astrocytes AND seizures OR epilepsy”, microglia AND seizures OR epilepsy”, “neuroinflammation AND epilepsy OR seizures”. We collected the outstanding information from the past 5 years (2019-2023), and also critical manuscripts in older years.

Section 2.1: Line 90: Please describe what “neuronal synapsis”, is. Otherwise, please check if this ought to be synapses i.e., formation of new synapses.

We do apologize for the mistake. We wanted to refer only to synapsis. We have changed the text accordingly.

Figure 1: Is the image of the astrocyte in the housekeeping or reactive state? Please can this be stated in Figure 1.

We have included the description of the housekeeping astrocyte in the legend of Figure 1.

Line 107 to 112: This paragraph comes across too vague especially when the authors state that “Due to the loss of their functional properties, reactive astrocytes are associated with the appearance of seizures.” The authors should provide further information how the loss of reactive astrocyte functionality leads to the appearance of seizures. Are these seizures categorised to a specific type i.e., focal, generalized? Are prominent to specific regions of the brain?

We have rephrased the sentence in the following way: “Due to the loss of their functional properties, reactive astrocytes fail in the maintenance of different homeostatic systems, which could lead to hyperexcitability. They have altered…”

Line 133: When describing the effect of adenosine metabolism in line 133, please can the authors clarify what is meant by ‘They” in “They increase”? Does “they” reflect changes in adenosine metabolism in reactive astrocytes?

We have rephrased the sentence in the following way: “Changes in adenosine metabolism are also produced in reactive astrocytes. These cells increase the expression…”

Line 160: more explanation should be added on the role of astrocytes as epileptogenic drivers and cite the relevant empirical evidence (as necessary).

We have added the following sentence: “… contributes to the appearance of hyperexcitability. Recently, an astrocytic basis for epilepsy has been proposed, and results both in animal models and in human samples indicate that astrocyte dysfunction can participate in hyperexcitation, neurotoxicity, and seizure spreading (Dossi, 2018). Perhaps, this is the reason why the European Commission of the International League Against Epilepsy (ILAE) recognized as a top research priority the work on the role that glia and inflammation may have on the development of seizures and epileptogenesis, and encouraged the identification of glial targets as a basis for the development of more specific anti-seizure medications (ASMs) (Baulac 2015).

Figure 2: Line 213 – please clarify what is meant by “neuronal dead cells”. From my understanding, the development and progression of epilepsy may not entirely be caused by neuronal cell death, for example, some paediatric neurodevelopmental disorders have co-occurring epilepsy with no neuronal cell death (neurodegeneration). Please also check whether the arrow(s) in Figure 2 ought to be bidirectional. As it stands, it is difficult to discern from Figure 2, how activated astrocytes lead to seizures. Please also described what ‘A’ and ‘M’ means.

We thank the reviewer for this comment. The drawing presented in Figure 2 is an oversimplification of the astrocyte-microglia crosstalk. To be more correct, we have changed the sentence in the following way: “…microglia are activated by the release of different components (e.g., ATP) from neuronal cells. Then, microglia induce the activation of astrocytes…”

We have also removed the term seizures from the drawing to focus only in the astrocyte-microglia crosstalk. In addition, we have included the reference to A: astrocytes, and M: microglia.

Line 234: If microglia can become activated after a seizure, please also explain in this section how reactive/activated microglia may also precipitate seizure(s). Can any information be provided whether reactive/activated microglia have a role in the pre-ictal stage and whether this might lead to the ictal phase and then the post-ictal phase including more deleterious post ictal aspects such as Post-ictal Generalized Electroencephalographic Suppression (PGES)? This can be linked in with the text on line 317 where the authors suggest that neuroinflammation may occur before the onset of seizures.

We have extended the relationship between microglia and epilepsy in the following way: “It has been assumed that microglia get activated rapidly after a seizure. Activated microglia, on the one hand, releases pro-inflammatory cytokines and toxic compounds (e.g., reactive oxygen species, ROS; nitric oxide, NO), and on the other hand, have dysfunctional phagocytosis and defective proteostasis, which is detrimental for neuronal function (see (51), (52) and (53) for review).

We are afraid that among the references we used, we did not find any information to answer the comment raised by the reviewer on the ictal stages of epileptic disorders.

Line 318: this is an interesting point and should be expanded upon. Can the authors shed light on any instances in which there is a susceptibility to dysregulated glial immune-inflammatory function that could predispose certain individuals to be at a higher risk of seizures?

As indicated in the text, astrocyte reactivity leads to changes in the homeostasis of glutamate transporters, which leads to hyperexcitability, excitotoxicity, and eventually, the appearance of seizures (ref. (25), (62)).

This hypothesis is supported by different reports in the literature as the ones indicated above. In this line of thinking, our group is interested in studying the molecular basis of Lafora disease, a severe form of progressive myoclonus epilepsy. In animal models of this disease, we observed a dysregulation of glutamate uptake, due to low levels of glutamate transporters at the level of the plasma membrane. Neuroinflammation is another trait of the disease and it becomes more severe as the animals age. These alterations lead to a higher sensitivity to the convulsant pentylenetetrazol (PTZ), as we describe in the Metformin part. Another example is type I progressive myoclonus epilepsy (Unverricht-Lundborg disease) characterized by the presence of activated microglia and neuroinflammation. These alterations also lead to the appearance of hyperexcitability and seizures (Tegelberg, 2012) (Okuneva, 2016). In these two cases, glial alterations and neuroinflammation precede the appearance of seizures.

Section 4: please say what Mabs and box are. I feel a table stating the main repurposed molecules and their putative roles would be useful here. It should also be clearly stated whether they would be useful for the management of epilepsy. Demarcation between studies done in animal models and humans should be clearly indicated.

Mabs stands for monoclonal antibodies; BoxA is the name of the compound. Following the reviewer’s suggestion we have included a Table with the repurposing drugs we mention in the text, where we indicate the primary indication, and whether the compound has been used in animal models or epileptic patients.

The potential limitation of these molecules should be also alluded too. For example, the failure of Fingolimod in Rett Syndrome, a paediatric neurodevelopmental disorder, where seizures are common.

Following the reviewer’s suggestion we have included this information in the text: “However, in the case of the Rett syndrome, although the administration of fingolimod was safe in children with this disorder, it did not provide supportive evidence for an effect on clinical, laboratory, and imaging measures in these patients” Naegelin et al Orphanet J Rare Dis. 2021 Jan 6;16(1):19. doi: 10.1186/s13023-020-01655-7.

Reviewer 2 Report

Comments and Suggestions for Authors

1.      “Neuroinflammation and epilepsy are two sides of the same coin” analogy is not appropriate, because neuroinflammation perturb and is central to many conditions, so can’t be a binary to epilepsy.

2.       Rather a visual representation e.g. a flow chart representation, of three categories i.e. inducers, sensors and effectors could be added to the introduction section as a more direct approach.

3.      Lines, 107-112; authors are advised to possibly include/mention the magnitude of seizers in clinical terms, which was predisposed by the lowered expression of Kir4.1 or any other indicated factors.

4.      Lines 120-127; the two mentioned functions of reactive astrocytes; are they mutually exclusive please clarify in the text?

5.      Lines 157-160; needs paraphrasing to clarify the drawn conclusion.

6.      Lines 186-190; Synaptic pruning is an event associated with the refining of neuronal circuits during developmental stage, so do authors are of opinion that any synaptic loss in epileptic-conditions (e.g. older epileptic patients) could be termed as synaptic-pruning, please clarify accordingly in the manuscript?

7.        Lines 188-89; in the cited literature (Ref. 39,40-41), please review the age groups of cohorts and update the manuscript accordingly.

8.      Fig.3.; Seizure-spikes doesn’t look the way it is depicted in the diagram, it is mostly a burst like episode, please rectify. Furthermore, it is appreciated if actual rage of seizure frequency is depicted preferably with a scale alongside (optional in a cartoon).

9.      For drug effects, e.g., Metformin affects in Alzheimer's, Parkinson's and in Huntington disease pathophysiology, authors are suggested to cite the original study reports or literature.

10.   All proposed drugs need more work to specify their potential roles in seizure, it is advisable that these drugs are reviewed in context to their mechanistic specificity and their respective subsequent role in seizures.

11.   All figures need further work, please do the needful.

1

Comments on the Quality of English Language

2.   Please at your convenience (if possible) simplify the language of your arguments further for wider audience, also look for typos.

Author Response

Reviewer 2:

  1. “Neuroinflammation and epilepsy are two sides of the same coin” analogy is not appropriate, because neuroinflammation perturb and is central to many conditions, so can’t be a binary to epilepsy.

We thank the reviewer for this comment. We have deleted this sentence from the abstract. Now it says: “Neuroinflammation and epilepsy are different entities but in some cases, they are so closely related that the activation of one of the pathologies…”

  1. Rather a visual representation e.g. a flow chart representation, of three categories i.e. inducers, sensors and effectors could be added to the introduction section as a more direct approach.

Following the reviewer’s suggestion we have added a flow chart representation of the inducers, sensors, and effectors.

  1. Lines, 107-112; authors are advised to possibly include/mention the magnitude of seizers in clinical terms, which was predisposed by the lowered expression of Kir4.1 or any other indicated factors.

We have rephrased the paragraph in the following way: “Due to the loss of their functional properties, reactive astrocytes fail in the maintenance of different homeostatic systems, which could lead to hyperexcitability. They have altered…”

  1. Lines 120-127; the two mentioned functions of reactive astrocytes; are they mutually exclusive please clarify in the text?

The alterations present in reactive astrocytes are not mutually exclusive. We have rephrased the paragraph in the following way: “… the release of cytokines such as TNF, IL-6, and IL-1b by reactive astrocytes aggravate excitotoxicity since, on the one hand, these molecules stimulate the release glutamate from these cells, and on the other hand, these cytokines increase the functionality of post-synaptic glutamate receptors…”

  1. Lines 157-160; needs paraphrasing to clarify the drawn conclusion.

Following the reviewer’s suggestion we have rephrased the paragraph in the following way: “… contributes to the appearance of hyperexcitability. Recently, an astrocytic basis for epilepsy has been proposed, and results both in animal models and in human samples indicate that astrocyte dysfunction can participate in hyperexcitation, neurotoxicity, and seizure spreading (Dossi, 2018). Perhaps, this is the reason why the European Commission of the International League Against Epilepsy (ILAE) recognized as a top research priority the work on the role that glia and inflammation may have on the development of seizures and epileptogenesis, and encouraged the identification of glial targets as a basis for the development of more specific anti-seizure medications (ASMs) (Baulac 2015).

  1. Lines 186-190; Synaptic pruning is an event associated with the refining of neuronal circuits during developmental stage, so do authors are of opinion that any synaptic loss in epileptic-conditions (e.g. older epileptic patients) could be termed as synaptic-pruning, please clarify accordingly in the manuscript?

In our opinion, synaptic pruning occurs not only during developmental stages but also in adulthood (Wong M, Guo D. Dendritic spine pathology in epilepsy:cause or consequence? Neuroscience. 2013;251:141–150). It is accepted that both astrocytes and microglia participate in neuronal plasticity and alterations in synaptic pruning may affect the neuronal circuits related to hyperexcitability, as we indicate in the manuscript.

  1. Lines 188-89; in the cited literature (Ref. 39,40-41), please review the age groups of cohorts and update the manuscript accordingly.

References 39, 40 and 41 are reviews on the topic. Some of the age groups cohorts in the original works are the following:

Böttcher C, Schlickeiser S, Sneeboer MAM, et al. Nat Neurosci. 2019;22(1):78-90. (from 23 to 80 years)

Sun FJ, Zhang CQ, Chen X, et al. Downregulation of CD47 and CD200 in patients with focal cortical dysplasia type IIb and tuberous sclerosis complex. J Neuroinflam. 2016;13(1):85. FCD IIb (range 1.8–9.5 years), 13 TSC (range 1.5–10 years) patients, and 6 control cases (range 1.5–11 years).

Gosselin D, Skola D, Coufal NG, et al. An environment-dependent transcriptional network specifies human microglia identity. Science. 2017;356(6344). Cohorts from 1 to 17 years of age.

  1. 3.; Seizure-spikes doesn’t look the way it is depicted in the diagram, it is mostly a burst like episode, please rectify. Furthermore, it is appreciated if actual rage of seizure frequency is depicted preferably with a scale alongside (optional in a cartoon).

We apologize for the oversimplification in drawing the seizure spikes. We have modified the figure including real seizure-spike diagrams.

  1. For drug effects, e.g., Metformin affects in Alzheimer's, Parkinson's and in Huntington disease pathophysiology, authors are suggested to cite the original study reports or literature.

Following the reviewer’s suggestion we now include the original study reports instead of the reviews:

AD: Koenig AM, Mechanic-Hamilton D, Xie SX, Combs MF, Cappola AR, Xie L, et al.. Effects of the insulin sensitizer metformin in Alzheimer Disease: pilot data from a randomized placebo-controlled crossover study. Alzheimer Dis Assoc Disord. (2017) 31:107–13.

PD: Brakedal B, Haugarvoll K, Tzoulis C. Simvastatin is associated with decreased risk of Parkinson disease. Ann Neurol. (2017) 81:329–30.

HD: Hervas D, Fornes-Ferrer V, Gomez-Escribano AP, Sequedo MD, Peiro C, Millan JM, et al.. Metformin intake associates with better cognitive function in patients with Huntington's disease. PLoS ONE (2017) 12:e0179283.

MS: Abdi et al., Metformin Therapy Attenuates Pro-inflammatory Microglia by Inhibiting NF-κB in Cuprizone Demyelinating Mouse Model of Multiple Sclerosis. Neurotox Res. 2021 Dec;39(6):1732-1746.

  1. All proposed drugs need more work to specify their potential roles in seizure, it is advisable that these drugs are reviewed in context to their mechanistic specificity and their respective subsequent role in seizures.

We have extended the description of the use of the drugs mentioned in this work including a Table with the repurposing drugs we mention in the text, where we indicate the primary indication, and whether the compound has been used in animal models or in epileptic patients

  1. All figures need further work, please do the needful.

We have modified some of the figures included in the work to make them more clear.

Round 2

Reviewer 1 Report

Comments and Suggestions for Authors

Thank you for revising the manuscript. One minor addition; please indicate in Section 2 Methodology that this was a narrative review (as you have stated in your cover letter).

Comments on the Quality of English Language

Acceptable

Author Response

Reviewer 1

Thank you for revising the manuscript. One minor addition; please indicate in Section 2 Methodology that this was a narrative review (as you have stated in your cover letter).

As suggested, we have indicated in Section 2 that our work was a narrative review.
